🔓 | **Open Peer Review** | Epidemiology | Research Article

# The influence of environmental factors on the detection and quantification of SARS-CoV-2 variants in dormitory wastewater at a primarily undergraduate institution

Chequita Brooks,[1,2] Sebrina Brooks,[3] Josie Beasley,[1] Jenna Valley,[1] Michael Opata,[1] Ece Karatan,[1] Rachel Bleich[1]

**ABSTRACT** Testing for the causative agent of coronavirus disease 2019 (COVID-19), severe acute respiratory syndrome coronavirus 2 (SARS-CoV-2), has been crucial in tracking disease spread and informing public health decisions. Wastewater-based epidemiology has helped to alleviate some of the strain of testing through broader, population-level surveillance, and has been applied widely on college campuses. However, questions remain about the impact of various sampling methods, target types, environmental factors, and infrastructure variables on SARS-CoV-2 detection. Here, we present a data set of over 800 wastewater samples that sheds light on the influence of a variety of these factors on SARS-CoV-2 quantification using droplet digital PCR (ddPCR) from building-specific sewage infrastructure. We consistently quantified a significantly higher number of copies of virus per liter for the target nucleocapsid 2 (N2) compared to nucleocapsid 1 (N1), regardless of the sampling method (grab vs composite). We further show some dormitory-specific differences in SARS-CoV-2 abundance, including correlations to dormitory population size. Environmental variables like precipitation and temperature show little to no impact on virus load, with the exception of higher temperatures for grab sample data. We observed significantly higher gene copy numbers of the Omicron variant than the Delta variant within ductile iron pipes but no difference in nucleocapsid abundance (N1 or N2) across the three different sewage pipe types in our data set. Our results indicate that contextual variables should be considered when interpreting wastewater-based epidemiological data.

**IMPORTANCE** Testing for severe acute respiratory syndrome coronavirus 2 (SARS-CoV-2), the causative agent of coronavirus disease 2019 (COVID-19), has been crucial in tracking the spread of the virus and informing public health decisions. SARS-CoV-2 viral RNA is shed by symptomatic and asymptomatic infected individuals, allowing its genetic material to be detected and quantified in wastewater. Here, we used wastewater-based epidemiology to measure SARS-CoV-2 viral RNA from several dormitories on the Appalachian State University campus and examined the impact of sampling methods, target types, environmental factors, and infrastructure variables on quantification. Changes in the quantification of SARS-CoV-2 were observed based on target type, as well as trends for the quantification of the Delta and Omicron variants by sampling method. These results highlight the value of applying the data-inquiry practices used in this study to better contextualize wastewater sampling results.

**KEYWORDS** wastewater-based epidemiology, SARS-CoV-2, droplet digital PCR

Address correspondence to Rachel Bleich, bleichrm@appstate.edu.

The authors declare no conflict of interest.

See the funding table on p. 14.

Wastewater-based epidemiology has been a tool used in detecting legal and illegal drug use since 2005, and, as of 2019, has been used for detecting environmental and anthropogenic stressors on humans (1). Since then, wastewater-based

10.1128/spectrum.02003-24 **1**

epidemiology has provided information on both active cases and the general prevalence of severe acute respiratory syndrome coronavirus 2 (SARS-CoV-2). Wastewater testing for SARS-CoV-2 has been shown to detect both asymptomatic and pre-symptomatic individuals, allowing for a more complete representation of the prevalence of disease (2). This method is not without drawbacks, however, as only approximately 50% of symptomatic individuals shed SARS-CoV-2 in fecal material (3). To our knowledge, there are no previous studies that show amounts/levels of viral shedding from asymptomatic individuals; therefore, the value for asymptomatic individuals is more difficult to quantify. Building-specific or near-to-source sampling in 2020 suggested the method is sensitive to detect infection if as little as approximately 0.25% (4, 5) to 3% (6) of the population tests positive via nasal swab. It is also important to note that the shedding rate varies between the parental strain and subsequent variants including Delta and Omicron (7). Once the virus leaves the host and enters wastewater infrastructure, a series of new factors influence the presence and detection of viral RNA .

Previous work has found that quantification of SARS-CoV-2 from wastewater is influenced by time (8), sample storage, handling, and collection methods (9–12), and RNA concentration methods (13, 14). Three methods of quantifying SARS-CoV-2 gene copy numbers have been employed: digital PCR (dPCR), droplet digital PCR (ddPCR), and quantitative PCR (qPCR). ddPCR has been suggested to outperform qPCR on low viral load samples (15–17). Many of these previous studies use a small number of samples, often targeted at wastewater treatment plants. It has also been suggested that the scale of the collection system (e.g., building level, wastewater treatment plant level) should inform which quantification method is used (18). Of particular interest to many scientists and administrators in academia was the wastewater tracking of SARS-CoV-2 from university dormitory populations (5, 10, 19–22), and sampling efforts on university campuses have varied in scale, source specificity, and methodology. Here, we present a data set of over 800 samples aimed at collating a variety of factors and their influence on SARS-CoV-2 quantification using ddPCR from building-specific sewage infrastructure.

A source of confusion in data reporting from SARS-CoV-2 tracking studies is the difference in definitions among researchers for different sample types. The definition of "grab sample" varies between manuscripts, referring to manually collected samples (5, 10, 19, 21) or samples collected hourly or sub-hourly by an autosampler and maintained separately (18, 23). Similarly, "composite" samples refer to manually collected and collated samples (24, 25) or samples collected by autosampler and stored as a composite for pickup and analysis (8, 9, 26). These discrepancies in the descriptions of the methodologies can make it difficult to compare studies. Here, we categorize a sample collected manually at two to three time points and subsequently combined as a "grab" sample. Our "composite" sample is defined as a sample collected in a single collection vessel by an autosampler over a period of 24 hours.

This study sought to quantify SARS-CoV-2 viral RNA from several dormitories on the Appalachian State University campus in Boone, North Carolina. Samples were collected twice per week from 1 March 2021 to 28 April 2022. This period of sampling represented three semesters over four seasons. During the course of this study, the variants of concern, Delta and Omicron, both became prevalent in the student body and were, therefore, quantified and tracked. The goal of this study was to further improve future wastewater-based epidemiological efforts by reporting correlations of our viral tracking with a variety of metadata including building occupancy, vaccination rates, weather, and sampling methods.

## MATERIALS AND METHODS

### Sample collection and processing

Samples were collected from 1 March 2021 to 28 April 2022 and cumulatively represent 14 different dormitory sewers. Grab samples were collected over three intervals of approximately 20 minutes at each site between 8 and 10 a.m. Additionally, five of

the sites used for grab sampling were also equipped with autosamplers (Global Water Instrumentation, Inc., College Station, TX, USA) programmed to collect every 2 hours over a 20 hour period beginning in August 2021. Immediately following collection, samples were transported to the laboratory in a cooler, and 200 mL aliquot of each composite sample (total volume of approximately 1 L) was heat pasteurized at 70°C for 30 minutes. A 200 mL "method blank" consisting of 1×PBS was also prepared, pasteurized, and processed alongside the samples. Following pasteurization, bovine respiratory coronavirus vaccine in the form of a modified live virus was added to a concentration of 1,300 copies/mL of wastewater sample (BCoV, BOVILIS Coronavirus, Merck Animal Health, Madison, NJ, USA), diluted per manufacturer instruction. The pH of each sample was then adjusted to 3.5 or lower using 10 N HCl for electronegative filtering (27). Immediately following, 4 mL of 1.25 M magnesium chloride was added to each 200 mL sample for a final concentration of 24.5 mM. Approximately 40 mL of the samples was filtered through a six-place vacuum manifold (EMD Millipore) using 0.45 µm mixed cellulose ester filters (MCE, MF-Millipore, Darmstadt, Germany). Filter paper was collected and stored in a 2 mL collection tube for RNA extraction.

## RNA extraction and reverse transcription

RNA was extracted from the filters using a modified version of the Qiagen RNeasy Plus Mini Kit (Qiagen, Hilden, Germany) protocol. RLT Buffer Plus (850 µL) was added to each 2 mL tube containing an entire filter and, after the filters were manually submerged in the buffer solution using the pipette tip, the samples were incubated for 10 minutes at room temperature. After briefly agitating the filter with a pipette tip, 550 µL of supernatant was transferred to a gDNA Eliminator spin column, and samples were centrifuged for 1 minute at 13,000 rpm. Five hundred fifty microliters of 70% molecular-grade ethanol was added to the flow-through, and the entire volume was transferred to an RNeasy Mini Spin column to be centrifuged for 1 minute at 13,000 rpm. Buffer RW1 (700 µL) was added to the spin column and centrifuged for 1 minute at 13,000 rpm. Buffer RPE (500 µL) was added to the spin column and centrifuged for 1 minute at 13,000 rpm. This step was repeated, with a second centrifugation for 2 minutes. The spin column was then "dried" with a 1 minute centrifugation at 13,000 rpm, followed by two elutions in the same 30 µL volume of RNase-free water centrifuged for 1 minute at 13,000 rpm. The samples were placed on ice and immediately used for cDNA synthesis. Negative controls for reverse transcription included a no reverse transcriptase and no-template control. cDNA synthesis was performed using SuperScript IV VILO Master Mix (Invitrogen, Waltham, MA, USA) in a 1:1 ratio with sample volume using the following cycling parameters: annealing for 10 minutes at 25°C, reverse transcription for 10 minutes at 50°C, enzyme inactivation for 5 minutes at 85°C, and a final holding step at 4°C. cDNA was stored at −80°C until ddPCR analysis.

## Droplet digital PCR

A ddPCR system (QX200, Bio-Rad Laboratories, Inc., Hercules, CA, USA) was used for quantification of viral RNA. Custom probes for nucleocapsid 1 and 2 (Table 1) were used to quantify SARS-CoV-2. Process controls were detected using PrimePCR Custom Assay 2019-nCoV_PMMV (pepper mild mottle virus) and PrimePCR Custom Assay 2019-nCoV_BCoV (bovine coronavirus). Master mixes were created using 2x ddPCR Supermix for Probes (No dUTP; Bio-Rad Laboratories, Inc., Hercules, CA, USA), nuclease-free water, and the ddPCR Mutation Assays (dMDS983315944 and dMDS761081950) before adding 5 µL of cDNA. SARS-CoV-2 genomic RNA (VR-1986D, ATCC, Manassas, VA, USA), the L452R "Delta" spike protein mutation (IDT gBlock, Table 2), and the N764K "Omicron" mutation (IDT gBlock, Table 2) were included as positive controls in a consistent concentration across plates where appropriate. Droplet generation and ddPCR were carried out following the manufacturer's instructions. PCR amplification included the following steps: 10 minute enzyme activation at 95°C, then 40 cycles of 30 seconds

**TABLE 1** Oligonucleotide sequences of forward and reverse primers and probes used for custom ddPCR assays

| Target, fluorophore | Forward sequence (5′ to 3′) | Reverse sequence (5′ to 3′) | Probe sequence (5′ to 3′) | Source |
|---|---|---|---|---|
| Nucleocapsid 1 (N1, FAM) | GACCCCAAAATCAGCGAAAT | TCTGGTTACTGCCAGTTGAATCTG | ACCCCGCATTACGTTTGGTGGACC | CDC (28) |
| Nucleocapsid 2 (N2, FAM) | TTACAAACATTGGCCGCAAA | GCGCGACATTCCGAAGAA | ACAATTTGCCCCCAGCGCTTCAG | CDC (28) |
| Pepper mild mottle virus (PMMV, HEX) | GAGTGGTTTGACCTTAACGTTGA | TTGTCGGTTGCAATGCAAGT | AGGCCTACCGAAGCAAATGTCG | Zhang et al. (29), Bio-Rad |
| Bovine coronavirus (BCoV, HEX) | CTGGAAGTTGGTGGAGTT | ATTATCGGCCTAACATACATC | CCTTCATATCTATACACATCAAGTTGTT | Decaro et al. (30) |

at 94°C and 1 minute at 55°C, followed by 10 minutes at 98°C for enzyme inactivation. Fluorescence signals were read using the Bio-Rad droplet reader.

## Target quantification

The data files (.qlp) generated by the Bio-Rad droplet reader were imported into the QX Manager Software, Standard Edition (Version 1.2.345.0909). Threshold values for droplet fluorescence were determined using standard deviation thresholding in the "2D Amplitude" function of the software. The standard deviation thresholding requires the user to select a point between the main positive and negative clusters observed in the program-defined channels 1 and 2 (FAM and HEX, respectively, in this study). Thresholds for all samples were set by first setting thresholds for the positive and negative process controls. The concentration of each target was quantified using the following assay probe pairs: N1 (FAM)/BCoV (HEX), N2 (FAM)/PMMV (HEX). The mutant assays for the detection of the Delta and Omicron variants are formulated with a pre-determined FAM/HEX probe pair. Due to variations in sample matrices common to wastewater samples, threshold values were manually verified and adjusted for each experimental sample to account for changes in background fluorescence. Once thresholds were applied, the concentration values determined in QX Manager were exported to a ".csv" file.

## Metadata

All ddPCR data and associated metadata are recorded in File S1. Nasal swab data for SARS-CoV-2 testing were obtained from the Appalachian State University COVID-19 Dashboard's cumulative and historical data (https://www.appstate.edu/go/coronavirus/reporting/dashboard-archive/index.php, accessed 28 December 2022 but no longer active), which were collected and provided on a weekly basis using on-campus health clinic data. The number of on-campus tests, number of positive tests, and the percent positive rate were downloaded for 2020-08-16 through 2022-04-25 (yr-mo-day). These values include results for all students, faculty, and staff tested at an on-campus clinic during this period. These data do not include any at-home rapid antigen tests or PCR tests performed at off-campus locations.

Dormitory occupancy and percent vaccination were collected on 21-09-17, 21-10-08, 21-10-25, 21-11-05, 21-11-19, 21-12-03, 22-01-07, 22-01-14, 22-01-21, 22-01-28, 22-02-04, 22-02-11, and 22-02-18. These values were applied to all wastewater sampling dates

**TABLE 2** Probe positive controls for ddPCR assays targeting the Delta and Omicron SARS-CoV-2 mutants

| Target | Positive control sequence |
|---|---|
| Delta L452R spike | CCCATGAGACATACAAAAAGGTAATGCCGCCTCGCTAGGTGAGCTACAGCTCGATTGTCACGTTAAGCTGGCCGCAAACTGGAAAGATTGCTGATTATAATT-ATAAATTACCAGATGATTTTACAGGCTGCGTTATAGCTTGGAATTCTAACAATCTTGATTCTCAAGGTTGGTGGTAATTATAATTACCGGTATAGATTGTTTAGGA-AGTCTAATCTCAAACCTTTTGAGAGAGATATTTCAACTGAAATCTATCAGGCCGGTAGCACACCTTGGTCTCGACTATACGCCCGTTTTCGGATC |
| Omicron N764K spike | CCCATGAGACATACAAAAAGGTAATGCCGCCTCGCTAGGTGAGCTACAGCTCGATTGTCACGTTAAGCTGGCCGACATCAGTAGATTGTACAATGTACATTT-GTGGTGATTCAACTGAATGCAGCAATCTTTTGTTGCAATATGGCAGTTTTTGTACACAATTAAAACGTGCTTTAACTGGAATAGCTGTTGAACAAGACAAAA-ACACCCAAGAAGTTTTTGCACAAGTCAAACAAATTTACAAAACACCACCGGTCTCGACTATACGCCCGTTTTCGGATC |

that fell between the report dates for occupancy and vaccination. For all sampling data collected prior to 21-09-17 or after 22-02-18, no vaccination rates were available and total occupancy was taken from the closest reported date (21-03-10, 21-09-01, or 22-02-01).

Temperature and precipitation data for each wastewater collection date were downloaded from the National Oceanic and Atmospheric Administration NOWData service (https://www.weather.gov/wrh/Climate?wfo=rnk, accessed 9 June 2022). These data were collected and collated by the Blacksburg, VA national weather service office for Boone, North Carolina. The average temperature and daily maximum precipitation are reported in our metadata. In instances where the precipitation was >0 and <0.01 inches, the value is recorded as trace (T).

## Statistics and analyses

All metadata were appended to the output file generated by QX Manager and opened in R (31) using the tidyverse package v. 1.3.0 (32) to create a data table. All samples were filtered so that only assays with greater than 10,000 accepted droplets were included in the analysis. The concentration value was then multiplied by the ddPCR reaction volume and divided by the volume of cDNA used in the reaction to calculate the target copies per microliter of the template. This value was multiplied by the total volume of the cDNA reaction divided by the starting volume of RNA to calculate the target copies per filter. The target copies per filter were then converted to target copies per liter by dividing by the filtered volume and multiplying by 1,000. The normality of target copies per liter was evaluated visually using the "ggdensity" and "ggqqplot" functions in the ggpubr R package v. 0.4.0 (33). The Shapiro-Wilk normality test was performed using the "shapiro.test" function in the dplyr R package v. 1.0.2 (34). Skew was tested using the "skewness" function in "moments" R package v. 0.14.1 (35). The target copies per liter were transformed for normality using a natural log transformation. Data were plotted using the R package ggplot2 v. 3.3.2 (36) with colorblind-friendly color palettes available using the ggthemes package v. 4.2.0 (37). Values were converted from decimals to percentages using the R "scales" package v. 1.1.1 (38).

## RESULTS

### Observations of SARS-CoV-2 and variants of concern

We can look across our collection timeline to observe spikes in SARS-CoV-2 abundance with the academic calendar. Samples of wastewater were collected from on-campus dormitories from March 2021 to April 2022 (Fig. S1). Archived samples from March 2021 to April 2022 were re-tested for the variant of concern Delta (Fig. S2). Archived samples from January 2022 to April 2022 were re-tested for the variant of concern Omicron (Fig. S3). The testing period for the variants of concern was chosen based on the first globally reported observations of each variant. The largest peaks in viral RNA count tended to happen in the weeks directly after major breaks in the calendar, such as summer and winter breaks (Fig. S1 to S3). The Delta variant was only detected during August and September of 2021, while the Omicron variant was detected across the entire spring semester of 2022 (Fig. S2 and S3).

### Sampling method and target type

To test whether there was a difference in the sampling methodology in detection of SARS-CoV-2 RNA, wastewater samples were collected by an automatic sampler set to sample 100 mL every 2 hours over 20 hours (composite) or as three hand-sampled 200 mL collections over an hour-long period in the 8–10 a.m. window of time (grab). This sampling comparison only included data represented by paired composite and grab samples from the same dormitory on the same day. There was no significant difference in viral RNA copies (Wilcoxon signed-rank test, $P = 0.16$) (Fig. 1) between grab or composite sampling observed among the 400 samples (200 composite samples; 200 grab samples)

tested between 17 August 2021 and 28 April 2022. There was no significant difference between grab or composite sampling for the detection of either of the targets N1 or N2 (Kruskal-Wallis, $P = 0.46$, 0.63, respectively) (Fig. 1), Delta (x = 107, Kruskal-Wallis, $P = 0.55$) (Fig. 2) or Omicron (x = 103, Kruskal-Wallis, $P = 0.053$) (Fig. 2) variants.

We next determined if there was a difference in target copies. To do this, a total of 3,800 ddPCR measurements were taken from our 813 wastewater samples and were quantified for 1,041 pairs of N1 (x = 1891) and N2 (x = 1909), including technical replicates between both grab and composite sampling techniques. Significantly higher target copies were detected for N2 over N1 (Wilcoxon signed-rank test, $P = 0.034$). Of the samples with detected SARS-CoV-2 RNA, 175 were quantified for Delta and 197 were quantified for Omicron. There was a significant difference in the natural log target copies

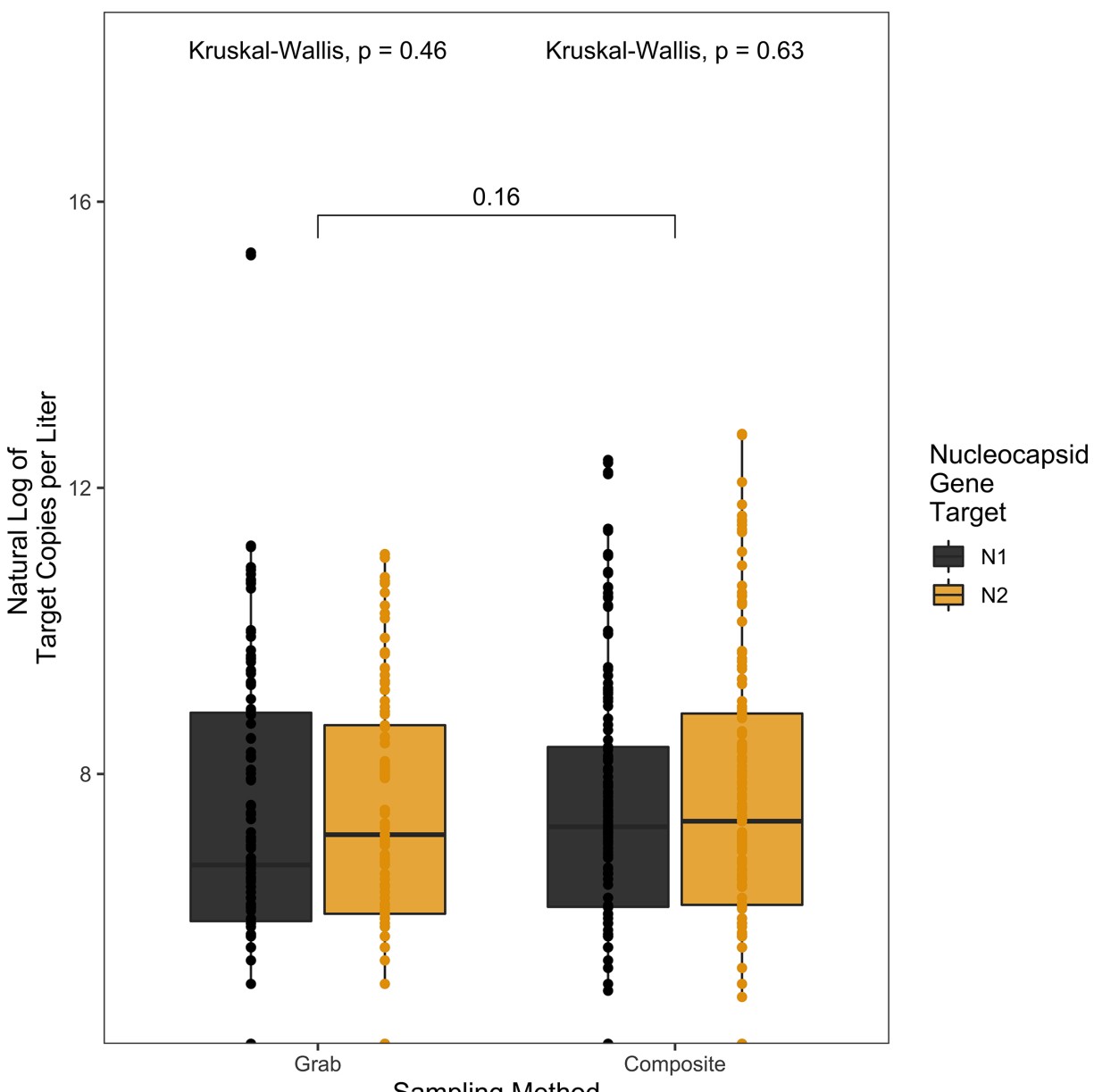

**FIG 1** Pairwise comparisons of the natural log of the target copies per liter by sampling method (grab, composite) and nucleocapsid gene target (N1 [black], N2 [yellow]). There was no significant difference between nucleocapsid gene targets within grab samples (Kruskal-Wallis, $P = 0.46$) or within composite samples (Kruskal-Wallis, $P = 0.63$). There was also no significant difference between grab and composite samples across both gene targets (Wilcoxon signed-rank test, $P = 0.16$).

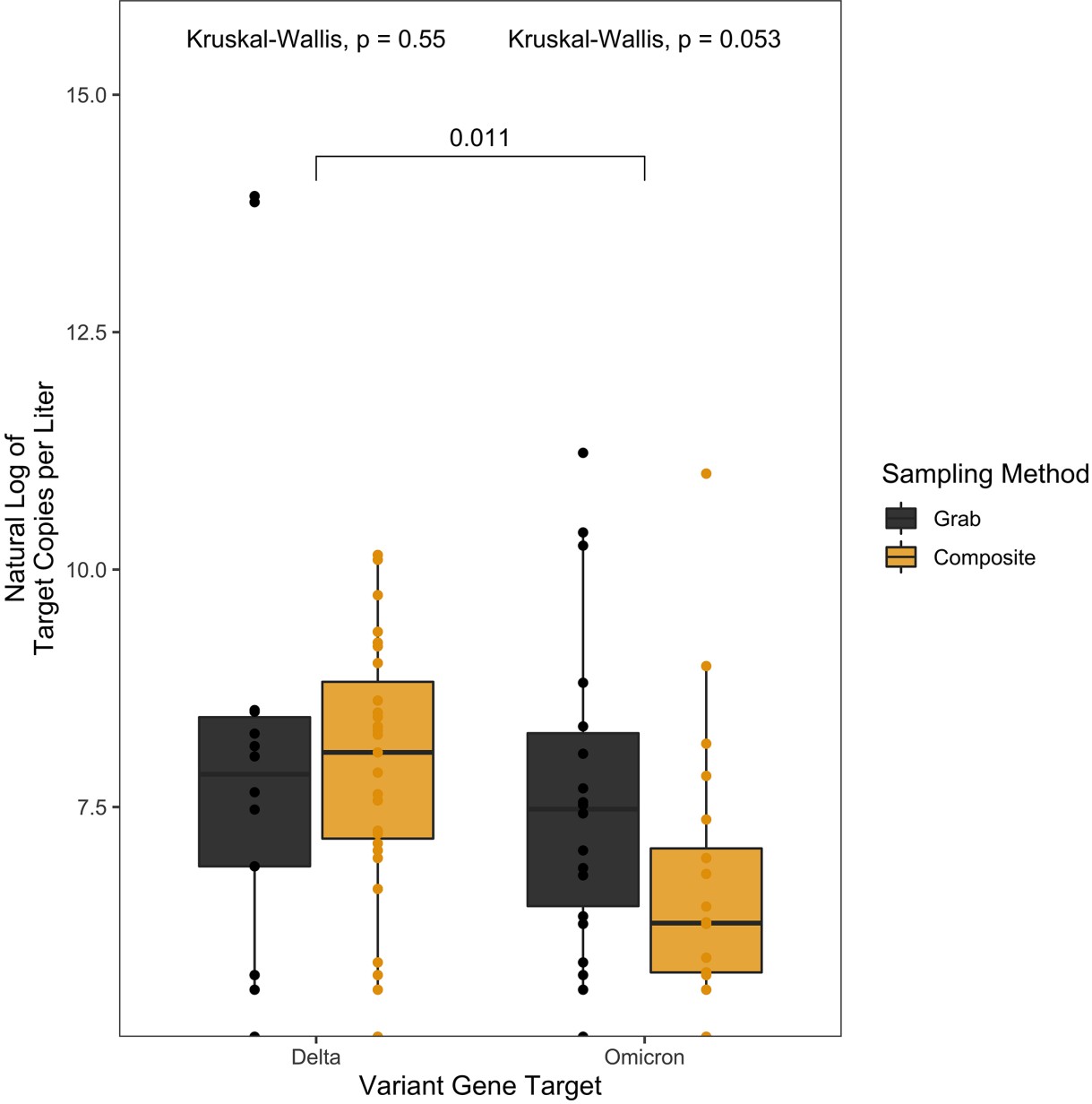

**FIG 2** Pairwise comparisons of the natural log of the target copies per liter by variant gene target (Delta, Omicron) and sampling method (grab [black], composite [yellow]). There was no significant difference between the sampling method for either variant of concern Delta (Kruskal-Wallis, *P* = 0.55) or Omicron (Kruskal-Wallis, *P* = 0.053). There was a significant difference between the detection of Delta and Omicron (Wilcoxon signed-rank test, *P* = 0.011).

per liter between Delta and Omicron (Wilcoxon, *P* = 0.011) for all SARS-CoV-2-positive samples tested (grab and composite pooled) (Fig. 2). When only comparing composite samples tested for the variants of SARS-CoV-2, there were significantly higher Delta copies per liter than Omicron observed (Wilcoxon, *P* = 0.0072). When comparing only grab samples tested for the variants of SARS-CoV-2, there were significantly higher Omicron copies per liter observed (Wilcoxon, *P* = 0.0057). However, these results are likely driven by outliers with extremely high copy numbers, even after normalization (Fig. 2).

## Dormitory population and vaccination rate

Fourteen dormitories were included in this data set, with dormitory populations ranging from 96 to 584 students. We observed a weak positive correlation between total dormitory population and the natural log of the target copies of N1 and N2 per liter (generalized linear model with Gaussian regressor, $R^2 = 0.017$, $P = 5.6e^{-5}$) (Fig. 3A). This relationship was not detected in the analysis of the variants ($R^2 = 0.018$, $P = 0.11$) (Fig. S4A).

From 17 September 2021 to 18 February 2022, the Appalachian State University tracked the number of coronavirus disease 2019 (COVID-19) vaccinated students (defined as having the first shot in any series). The percent vaccination rate, defined as the number of vaccinated students out of the total population of students in each dorm, ranged from 70.29% to 86.87%. We observed a weak positive correlation between

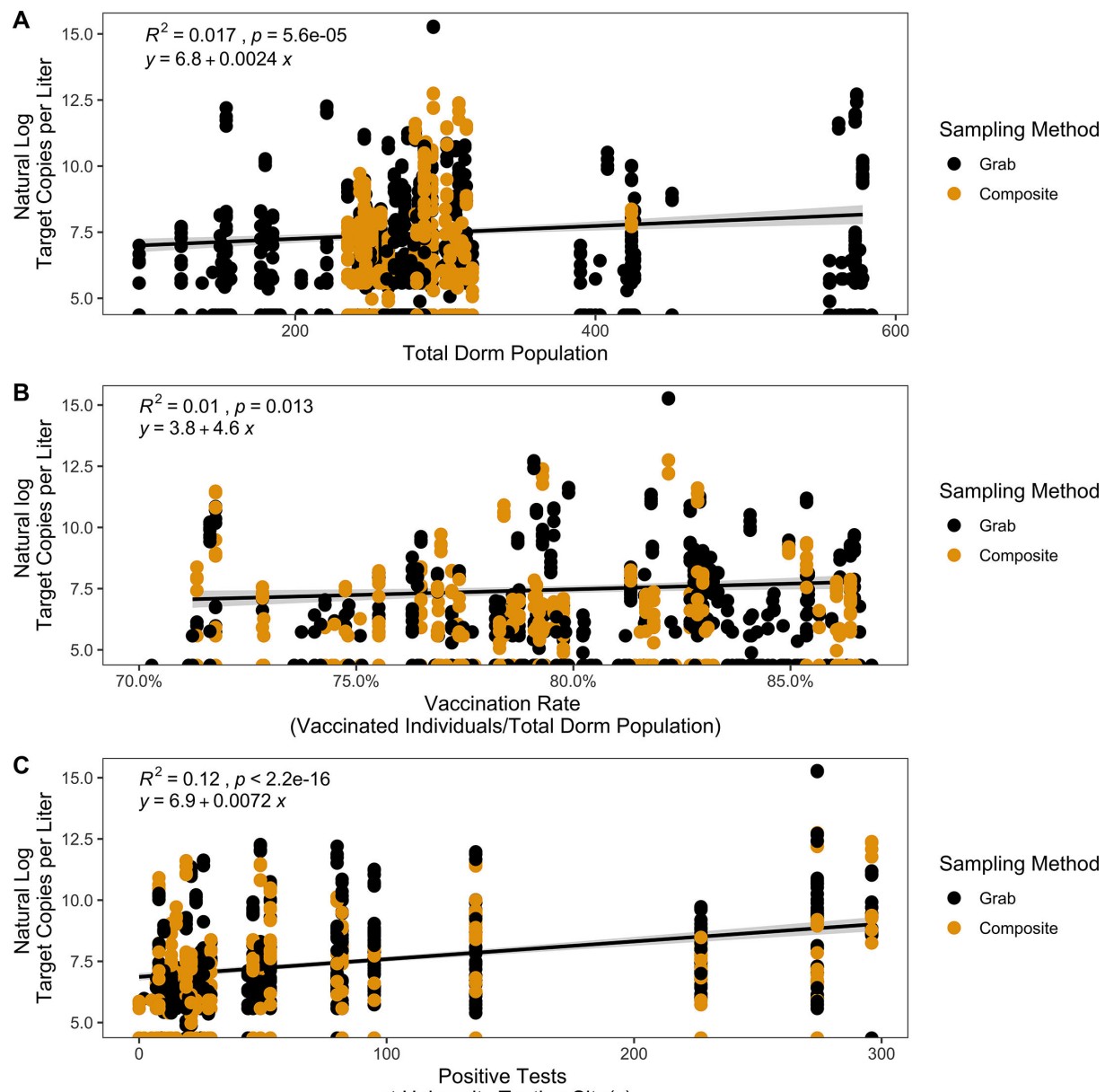

**FIG 3** Correlations between natural log target copies per liter for N1 and N2 gene targets for SARS-CoV-2 from dormitory wastewater samples and (A) total dormitory population, (B) vaccination rate per dormitory, and (C) total positive test results across university testing site(s). Correlations were tested using generalized linear models with Gaussian regressors.

percent vaccination and the natural log of the target copies per liter (generalized linear model with a Gaussian regressor, $R^2 = 0.01$, $P = 0.013$) (Fig. 3B). This relationship is not detected for the variants ($R^2 = 2.4e^{-7}$, $P = 1$) (Fig. S4B). From 16 August 2021 to 25 April 2022, the Appalachian State University tracked the number of positive SARS-CoV-2 nasal swab tests on a weekly basis for all faculty, staff, and students tested at a university-sponsored testing location. We observed a weak positive correlation between positive SARS-CoV-2 nasal swab test results and the natural log of the target copies per liter (generalized linear model with a Gaussian regressor, $R^2 = 0.12$, $P < 2.2e^{-16}$) (Fig. 3C).

Using a Wilcoxon rank-sum test, we performed pairwise comparisons of the mean natural log target copies per liter of N1 for each dormitory (Fig. 4). One dormitory, H, was significantly different from all dormitories except M and N. Dormitories M and N were not significantly different from any of the other dormitories sampled.

## Precipitation and temperature

Over the course of sampling, precipitation varied from 0 to 3 inches, and average daily air temperature ranged from −5.83°C to 21.67°C in Boone, North Carolina. We observed no correlation between precipitation and the natural log of target copies of N1 and N2 per liter ($R^2 = 4.3e^{-5}$, $P = 0.85$) (Fig. 5A). When examined independently, there was no correlation for either grab ($R^2 = 0.00027$, $P = 0.69$) or composite ($R^2 = 1.7e^{-5}$, $P = 0.95$) samples with precipitation. However, there was a weak but significant negative correlation ($R^2 = 0.03$, $P = 0.025$) (Fig. 5B) between precipitation and the highest natural log of target copies per liter ($9 \leq n \leq 17$, $n = 190$) of all dormitories represented in the data set. When examining the subset of samples with the highest target copies per liter from each dormitory, the maximum precipitation observed in that subset was lower than

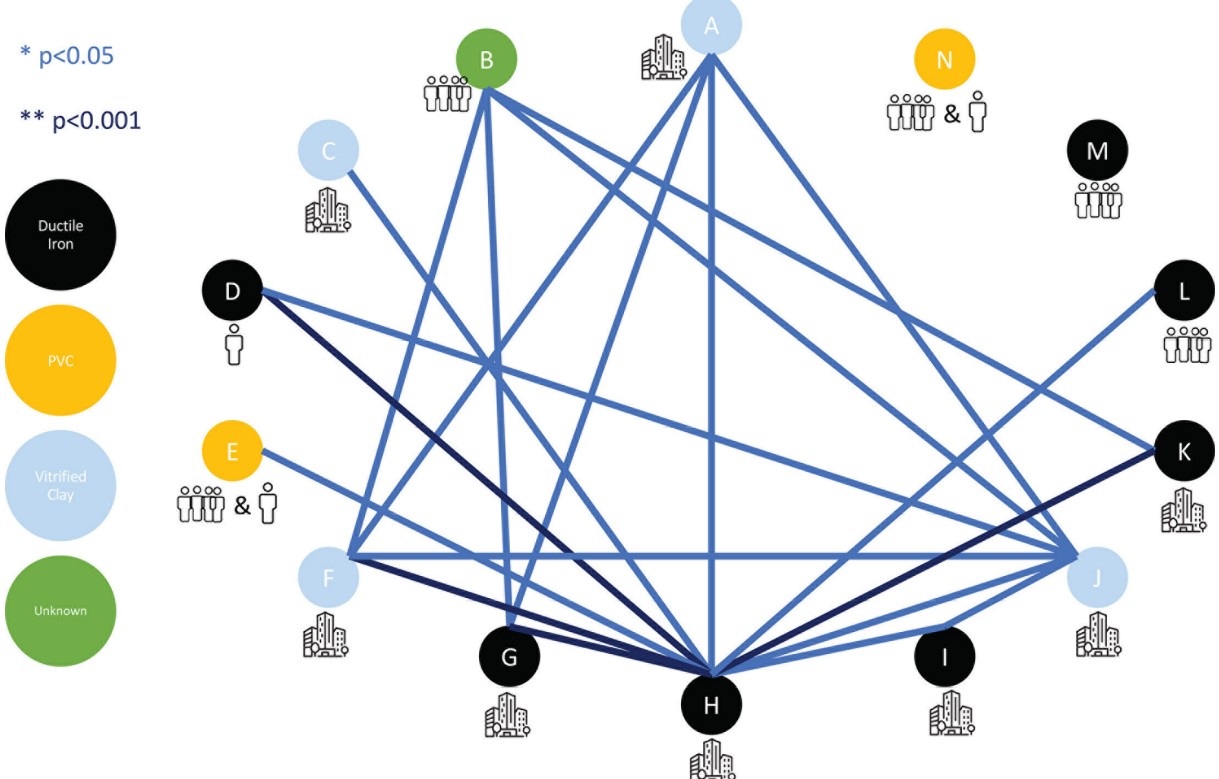

**FIG 4** Using a Wilcoxon rank-sum test, the mean log target copies of N1 per liter were compared pairwise across all 14 of the sampled dormitories. A *P* value < 0.05 is displayed in light blue, and a *P* value < 0.001 is displayed in dark blue. Each of the dormitories is coded with a letter (A–N), and the pipe type is color coded as ductile iron (black), polyvinyl chloride (PVC, yellow), vitrified clay (light blue), and unknown (green). The bathroom types are displayed below each color-coded letter, as either a single-unit bathroom (one-person outline), a shared-unit bathroom (four-person outlines), or a hall communal bathroom (building outline).

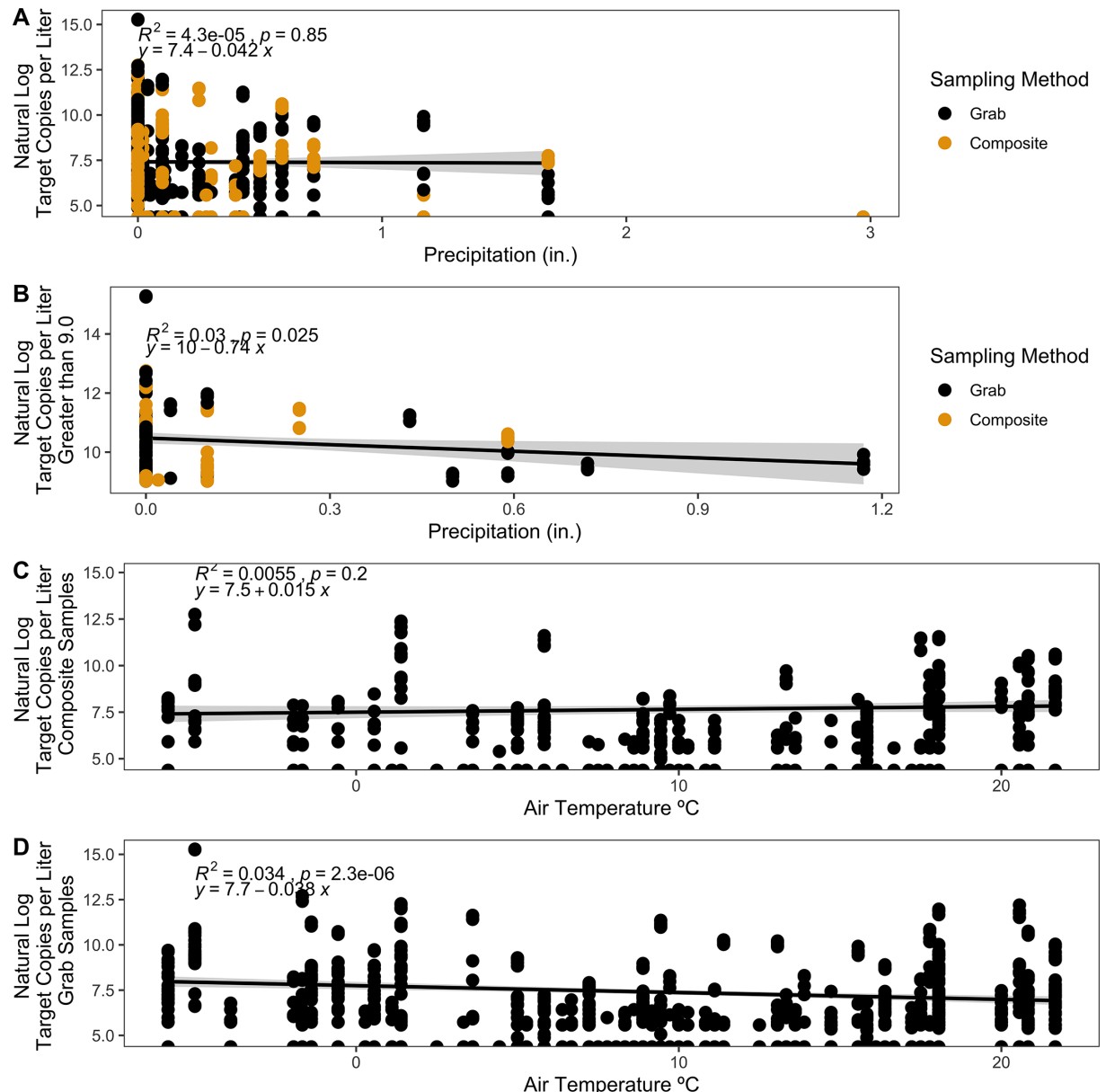

**FIG 5** Correlations between natural log target copies per liter for N1 and N2 gene targets for SARS-CoV-2 from dormitory wastewater samples and (A) precipitation (inches) across all grab (black) and composite (yellow) samples using a generalized linear model with Gaussian regressor. (B) The correlation between natural log target copies per liter ≥9 threshold and precipitation (inches) across all grab (black) and composite (yellow) samples using a generalized linear model with Gaussian regressor. (C) Natural log target copies per liter of composite samples by air temperature (°C) using a generalized linear model with Gaussian regressor. (D) Natural log target copies per liter of grab samples by air temperature (°C) using a generalized linear model with Gaussian regressor.

the maximum precipitation observed overall, with the highest precipitation of 1.2 inches only including datapoints from dorm N after quality control.

We observed no correlation between air temperature and the natural log of target copies of N1 and N2 per liter for composite samples ($R^2 = 0.0055$, $P = 0.2$) (Fig. 5C). However, we did observe a weak negative correlation with higher temperatures and grab samples ($R^2 = 0.034$, $P = 2.3e^{-6}$) (Fig. 5D).

## Sewage infrastructure

On the Appalachian State University campus, three different types of pipe infrastructure were available for wastewater sampling: ductile iron, polyvinyl chloride (PVC), and

vitrified clay. One of the dormitories sampled (B) had an unknown pipe type and was dropped from this analysis. We observed no significant difference in quantification between any of the pipe types sampled (Kruskal-Wallis, $P = 0.15$) for observed natural log of target copies of N1 and N2 per liter. We also observed no significant difference of pipe type on the observed natural log of target copies of Delta or Omicron per liter (Kruskal-Wallis, $P = 0.37$). We did observe a significantly higher quantification of Omicron than Delta within the ductile iron pipes (Kruskal-Wallis, $P = 0.00075$).

## DISCUSSION

### Sampling methodology

The efficacy of composite versus grab sampling remains unclear in the literature. Previous studies reporting sampling from wastewater treatment plants have also found no significant differences between grab and composite sampling techniques (24, 26, 39). Other studies reported a high number of non-detects in grab sampling from building-specific samples (18), while composites (pooled grab samples) were found to be significantly better at mid-point catchments (25). The time of day that grab sampling is performed has been found to influence the likelihood of non-detects due to factors such as flow rate (24, 40). However, our results indicate that, for the purposes of dormitory testing, using a methodology of morning sampling (between 8 and 10 a.m.) is sufficient to detect SARS-CoV-2 from grab sampling. This finding is of value to universities or governments that have limited infrastructure and funding, as the cost of purchasing autosamplers can be avoided, and as both grab and composite sampling techniques still require significant and similar amounts of personnel time. Furthermore, we found that many of our dormitories could not be monitored using autosamplers, as autosampler hoses frequently turned around, especially at peak flow times, leading to undersampling. Hence, it is important to consider building-specific wastewater infrastructure characteristics when devising sampling techniques.

### Primer probes (N1 vs N2)

The relative detection sensitivity of primer-probe sets for N1 and N2 remains somewhat ambiguous in the literature. In a comparison of qPCR results from clinical samples, the CDC's 2019-nCoV_N1 set was found to have a higher PCR efficiency and analytical sensitivity than the 2019-nCoV_N2 set (41, 42). In a study of dormitory-specific sampling using both RT-qPCR and RT-ddPCR, there was no significant difference in detection between the two primer-probe sets (19). However, in a study of wastewater effluent from three hospitals, it was found that the N1 primer-probe set was more sensitive (6) and there was a greater percentage of N1 primer-probe positive samples in eight wastewater samples (43). In a study of nine neighborhood sewer sheds, it was found that N1 and N2 were highly correlated, except in one instance where N2 had a much higher concentration (44). The variation in the correlation of primer-probe sets between studies suggests that there may be background influences in the wastewater. Future studies could include an analysis of the chemical makeup of the wastewater. More recent work has also demonstrated that mutations in the nucleocapsid sequence led to an underestimation of the N1 probe region after July 2022 using qPCR (45); however, the authors report that digital PCR is robust against mutations. Our samples had a significantly higher quantity of N2 over N1 after log normalization, which may be attributable to a mutation that the ddPCR method was sensitive to, or a difference in RNA stability under certain wastewater matrix conditions that we cannot predict from our available data set.

### Variants of concern

We detected a significant difference between viral RNA quantities between Delta and Omicron from the samples tested. Our results indicate that more copies of Delta were observed in composite samples and more copies of Omicron were observed from grab

samples. In a study of wastewater reclamation facilities in Florida and Arizona, it was found that in three of the samples, two of which were composite samples, there was a significantly higher detection of the Delta variant than the Omicron variant (7). It is unclear if this difference in detection is owed to differences in persistence outside the host between the two variants. A potential explanation could be that the Delta variant is more persistent within the conditions of the autosampler or is less susceptible to degradation in wastewater matrices, and so is collected to greater concentration over time. In a study of ancestral SARS-CoV-2 and variants of concern Alpha, Beta, and Omicron, it was observed that Omicron was less stable in liquid nasal mucus than the ancestral strain (46). To our knowledge, there are no comparisons in wastewater matrices of strain persistence time. We therefore recommend that differences in persistence in the environment between microorganisms is worth exploring when considering wastewater collection-based comparisons of future infectious agents.

## Dormitory population size and vaccination rate

We observed a positive correlation between dormitory population size and SARS-CoV-2 detection that is consistent with previous reports from wastewater catchments servicing populations of varying sizes (47). While it is not surprising that an increase in population increases the likelihood that SARS-CoV-2 would be detected from wastewater samples, we did not observe the same trend in the quantified variants of concern, possibly due to a lower sample size. However, the observed relationship with SARS-CoV-2 is very weak, so it would seem unlikely that population size, at least within this scale, would lead to an increase in type I or type II error in similar wastewater-based epidemiology studies. There was a weak positive correlation observed between vaccination rate and SARS-CoV-2 detection. However, it is unlikely that vaccination of the student population influenced our quantification, as none of the World Health Organization (WHO) approved vaccinations have been found to pose a risk for false positives due to vaccine-derived viral shedding (22, 48). Therefore, the weak positive correlation could be due to other factors such as overall frequency of infection or relaxing of other preventative measures. This hypothesis is supported by the positive correlation between the detected target copies and positive test results at university-sponsored testing centers.

## Environmental conditions

We observed no correlation between precipitation and the natural log of target copies of SARS-CoV-2 (N1 and N2) per liter; however, we did observe a significant negative correlation between precipitation and the highest observed copies per liter. Increased precipitation has been previously correlated with a decrease in detectable SARS-CoV-2, or complete loss of signal, both at site level and in wastewater treatment plants (20, 49). While that observation was primarily driven by sampling from combined sewer systems (incorporating storm drains into wastewater treatment pipelines), our study sampled from direct sewer lines. It may therefore be pertinent to consider the effects of precipitation on sample integrity, depending on the choice of sampling location. Water entering sewer systems that are not part of the sanitary flow is classified as either inflow or infiltration. Inflow includes leaking manhole covers and street wash water, and infiltration includes defective (leaky) pipes, pipe joints, or connections (50). Inflow and infiltration increase the flow rate of wastewater and dilute sanitary flow (50), both factors that would decrease the observed copy numbers for targets such as SARS-CoV-2. The samples collected for this study were from direct sewer lines, with no connections prior to the first manhole, which was used to access building-specific samples. Samples collected further downstream in the sewage infrastructure have a higher likelihood of being impacted by precipitation, and precipitation should be incorporated into data sets from those locations as an explanatory variable.

We observed no correlation between air temperature and the observed quantities of SARS-CoV-2 RNA in our samples. Air temperature has previously been found to

be important for modeling wastewater SARS-CoV-2 quantification from wastewater treatment plants (51, 52). Our results indicate that it would prove beneficial for researchers to develop compensation models for both temperature and precipitation when quantifying RNA viruses in wastewater, especially between studies. As RNA is very sensitive, especially to temperature, it is reasonable that some influence would be observed on SARS-CoV-2 levels, especially as sampling locations move further downstream from the point of source.

In this study, we also explored pipe type as a potential explanatory variable and did not find a significant effect of sewage infrastructure to SARS-CoV-2 quantification. The choice of piping between ductile iron and PVC has been shown to lead to differentiation between biofilm community members (53), but no work to-date has been carried out on the suspended organisms that would likely travel to wastewater treatment plants. Differences in the microbial community of wastewater treatment plant influents have also been observed that were not wholly attributable to latitude (i.e., temperature) (54). While overall SARS-CoV-2 quantification was not affected, we did observe a significantly higher quantification of Omicron than Delta variants of concern in ductile iron pipes. These results indicate that there is some small interaction between the viral particles and detectable RNA and their piping environment, which may be worth exploring in future studies. It is unclear, due to the lack of literature on the subject, if samples further from the point of source would have a greater observable impact from sewage infrastructure.

## Inter-dormitory differences

In a comparison of results between individual dormitories, we found that one dormitory, H, was significantly different from all but two of the other dormitories. We explored potential correlations with explanatory variables including bathroom type per dormitory (i.e., whether individual, shared by suite, or shared by hall), pipe type used for each dormitory, and dormitory population size, but we did not observe any differentiating factors between dormitories that may have led to these results. Our study did not measure chemical characteristics of the wastewater samples, such as chloride ($Cl^-$), which could have been used as a proxy for antiviral cleaning behaviors, or other wastewater characteristics such as microplastic concentration (55). We hypothesize that the differences in wastewater matrices between Dorm H and the other dormitories likely contributed to the observed result.

## Conclusions

This study comprises one of the largest sample data sets for SARS-CoV-2 quantification from building-level samples. Using a variety of metadata, we were able to observe a significant increase in the quantification of SARS-CoV-2 using the probe for N2 over N1, as well as trends for the quantification of the Delta and Omicron variants by sampling method. These results are especially informative in the context of future sampling efforts for wastewater-based epidemiology, as not many studies published since 2020 considered environmental variables that could be of great importance for data interpretation. While this study is geographically bound to a single college campus, we see a great value in applying the data-inquiry practices used in this study to future studies in order to better contextualize wastewater sampling results.

## ACKNOWLEDGMENTS

We would like to acknowledge Eric Johnson and Bio-Rad for their willingness to assist with establishing protocols, interpreting results, and building our understanding of the droplet digital PCR chemistry. We also acknowledge Arthur Frampton and Jacob Kazenelson from the University of North Carolina at Wilmington, and Joseph Bisesi from the University of Florida, for consultations about supplies, equipment, protocols, and workflows. We thank Rachel Noble and her lab members Thomas Clerkin and Denene Blackwood from the University of North Carolina at Chapel Hill for sharing their expertise

that helped us implement wastewater-based epidemiology at Appalachian State. Thank you to our undergraduate researchers, Tyler Perryman, Seth Leonard, and McKenna Beachum, who helped to process water samples. A huge thank you to Eric Greer and his team in Facilities Operations who collected the samples and taught us so much about sewage infrastructure. We would like to acknowledge Dr. Alex Howard and Appalachian State Student Affairs for helpful discussions and providing data.

We would like to thank Appalachian State University for start-up funding (R.B.) as well as additional financial support (R.B., M.O., J.V.) offered for this project. This project was supported by the North Carolina Policy Collaboratory (J.V., E.K., R.B., C.B.) at the University of North Carolina at Chapel Hill with funding from the North Carolina Coronavirus Relief Fund, established and appropriated by the North Carolina General Assembly.

J.V., M.O., E.K., and R.B. contributed to the supervision for the project. C.B., J.V., M.O., E.K., and R.B. contributed to funding acquisition. C.B., J.V., M.O., E.K., and R.B. contributed to the conceptualization of the project. C.B., S.B., J.B., J.V., M.O., and R.B. contributed to the investigation, formal analysis, and data curation for this project. C.B., S.B., and J.B. contributed to the writing of the original draft of the manuscript. C.B., E.K., J.V., M.O., and R.B. contributed to the review and editing of the manuscript.

## AUTHOR AFFILIATIONS

[1]Department of Biology, Appalachian State University, Boone, North Carolina, USA
[2]Louisiana Universities Marine Consortium, Chauvin, Louisiana, USA
[3]Department of Biology, University of North Carolina at Wilmington, Wilmington, North Carolina, USA

## AUTHOR ORCIDs

Rachel Bleich  http://orcid.org/0000-0002-8170-483X

## FUNDING

| Funder | Grant(s) | Author(s) |
| --- | --- | --- |
| UNC \| UNC-CH \| North Carolina Policy Collaboratory | 5125606 | Jenna Valley |
| | | Michael Opata |
| | | Ece Karatan |
| | | Rachel Bleich |
| Appalachian State University (ASU) | | Jenna Valley |
| | | Michael Opata |
| | | Rachel Bleich |

## AUTHOR CONTRIBUTIONS

Chequita Brooks, Conceptualization, Data curation, Formal analysis, Funding acquisition, Investigation, Writing – original draft, Writing – review and editing | Sebrina Brooks, Data curation, Formal analysis, Investigation, Writing – original draft | Josie Beasley, Data curation, Formal analysis, Investigation, Writing – original draft | Jenna Valley, Conceptualization, Data curation, Formal analysis, Funding acquisition, Investigation, Supervision, Writing – review and editing | Michael Opata, Conceptualization, Data curation, Formal analysis, Funding acquisition, Investigation, Supervision, Writing – review and editing | Ece Karatan, Conceptualization, Funding acquisition, Supervision, Writing – review and editing | Rachel Bleich, Conceptualization, Data curation, Formal analysis, Funding acquisition, Investigation, Supervision, Writing – review and editing

## ADDITIONAL FILES

The following material is available online.

## Supplemental Material

**File S1 (Spectrum02003-24-s0001.xls).** Full sample data set for SARS-CoV-2 abundance with metadata.

**Supplemental figures (Spectrum02003-24-s0002.pdf).** Fig. S1 to S4.

## Open Peer Review

**PEER REVIEW HISTORY (review-history.pdf).** An accounting of the reviewer comments and feedback.

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
