## [Reviewer comments · Microbiology Spectrum]

Microbiology Spectrum

The influence of environmental factors on the detection and quantification of SARS-CoV-2 Variants in Dormitory Wastewater at a Primarily Undergraduate Institution

Chequita Brooks, Sebrina Brooks, Josie Beasley, Jenna Valley, Michael Opata, Ece Karatan, and Rachel Bleich

Corresponding Author(s): Rachel Bleich, Appalachian State University Department of Biology

Review Timeline:

Submission Date:	August 22, 2024
Editorial Decision:	October 10, 2024
Revision Received:	December 10, 2024
Accepted:	December 12, 2024

Editor: JJ Miranda

Reviewer(s): The reviewers have opted to remain anonymous.

Transaction Report:

DOI: <https://doi.org/10.1128/spectrum.02003-24>

Re: Spectrum02003-24 (The influence of environmental factors on the detection and quantification of SARS-CoV-2 Variants in Dormitory Wastewater at a Primarily Undergraduate Institution)

Dear Dr. Rachel Bleich:

Thank you for the privilege of reviewing your work. Below you will find my comments, instructions from the Spectrum editorial office, and the reviewer comments.

Thank you for your submission examining technical aspects of SARS-CoV-2 detection in wastewater. Two expert reviewers have provided constructive feedback. In your revised manuscript, please pay particular attention to questions regarding the rigorous interpretation of the data.

Revision Guidelines

Sincerely,
JJ Miranda
Editor
Microbiology Spectrum

Reviewer #1 (Comments for the Author):

The manuscript presents data obtained from SARS-CoV-2 wastewater surveillance from dormitories on a university campus. Various characteristics of sampling strategy, RNA detection methodology, and viral (variant), environmental (e.g. pipe type) and

climate (e.g. precipitation) differences are examined to identify correlates for successful sampling. Overall, the authors found differences in SARS-CoV-2 RNA detection based upon nucleocapsid gene primer pairs (N1 vs N2); and by variant (Delta vs Omicron) by sampling method (grab vs. composite). These results could potentially inform future wastewater surveillance efforts.

Overall, the manuscript is well-written and clear. A major weakness is that the results are almost entirely presented textually, making it difficult to analyze directly. The major statistically significant findings should be represented graphically in figures. Additionally, most of the correlations observed were very weak, though perhaps statistically significant, making it unclear how relevant these observations are for informing future efforts.

Additional comments:

The authors refer to detection of "viral particles" - detection of RNA may be from particles but could also be from cellular or other debris; thus, it should always be stated that "viral RNA" was detected, not viral particles (or viral load).

It is also of note that Dorm H was significantly different from all of the other dorms, but no commentary was provided as to why this might occur - is it significantly smaller? had fewer cases? I understand why identifying information may not be presented, but information as to the percentage of positive samples by dorm, the timing of those samples (Delta vs Omicron), etc., would help to provide more information on possible explanations for the observed differences.

Reviewer #2 (Comments for the Author):

The authors note at one point in the paper no significant differences between N1 and N2 detection yet concluded that N2 was detected in significantly higher quantities, this should be cleared up.

Reviewer Comments for: The influence of environmental factors on the detection and quantification of SARS-CoV-2 Variants in Dormitory Wastewater at a Primarily Undergraduate Institution

This article addresses an important topic in public health surveillance, consideration of environmental factors in wastewater analysis. The paper understandable, however it would be better if the results and discussion were separate sections.

Authors here discussed results showing significantly higher quantifications for N2 versus N1. Other studies have shown this may be due to mutations in SARs-CoV-2. This should be discussed/mentioned in the current study.

Reviewers also stated that composite samples showed significantly higher delta copies while grab samples showed significantly higher omicron copies. Do researchers have a theory/explanation for this? Were the composite samples and grab samples taken during the same time frame? Is it possible that as time passed one variant became more dominant, thus skewing rates?

Did authors detect differences between semesters? As students left for school breaks, it's possible there was more exposure to one variant over another, particularly as time went on.

Response to Reviewers Running Title: Dorm Wastewater Testing

Reviewer #1 (Comments for the Author):

The manuscript presents data obtained from SARS-CoV-2 wastewater surveillance from dormitories on a university campus. Various characteristics of sampling strategy, RNA detection methodology, and viral (variant), environmental (e.g. pipe type) and climate (e.g. precipitation) differences are examined to identify correlates for successful sampling. Overall, the authors found differences in SARS-CoV-2 RNA detection based upon nucleocapsid gene primer pairs (N1 vs N2); and by variant (Delta vs Omicron) by sampling method (grab vs. composite). These results could potentially inform future wastewater surveillance efforts.

Overall, the manuscript is well-written and clear. **A major weakness is that the results are almost entirely presented textually, making it difficult to analyze directly.** The major statistically significant findings should be represented graphically in figures. **Additionally, most of the correlations observed were very weak, though perhaps statistically significant, making it unclear how relevant these observations are for informing future efforts.**

Response 1.1: We thank the reviewer for pointing out that it would be beneficial to readers to have the figures to reference for their own interpretation. We have added four figures to the main text and four figures to a supplementary figures file to address this weakness of the manuscript.

Response 1.2: We respectfully disagree with the reviewer that this work is not relevant to future wastewater sampling efforts. We have carefully indicated future considerations and missing data in the literature that are intended to inform future efforts and improve outcomes. While many of our statistical analyses had weak correlations, these also inform the data collection of future efforts and will assist in the collection of more holistic data sets that better represent the wastewater matrices, something currently missing in the literature. While we regret that we did not perform that work, we hope that our commentary will help to guide others.

Additional comments:

The authors refer to detection of "viral particles" - detection of RNA may be from particles but could also be from cellular or other debris; thus, it should always be stated that "viral RNA" was detected, not viral particles (or viral load).

Response 1.3: We thank the reviewers for pointing out this misnomer and have fixed the instances in the manuscript where we suggested we were measuring viral particles to say "viral RNA."

It is also of note that Dorm H was significantly different from all of the other dorms, but no commentary was provided as to why this might occur - is it significantly smaller? had fewer cases? I understand why identifying information may not be presented, but information as to the percentage of positive samples by dorm, the timing of those samples (Delta vs Omicron), etc., would help to provide more information on possible explanations for the observed differences.

Response 1.4: We have expanded on our hypotheses regarding why Dormitory H had a significantly different result from the other dormitories in the Discussion section. We appreciate the reviewer's suggestions that percentage of positive samples by dormitory would aid in the interpretation of this data set, however, we do not have access to that data. All dormitories were sampled across the same period of time and would have had an equal likelihood of delta and omicron infections.

Reviewer #2 (Comments for the Author):

The authors note at one point in the paper no significant differences between N1 and N2 detection yet concluded that N2 was detected in significantly higher quantities, this should be cleared up.

Response 2.1: We apologize to the reviewer for the confusion. We mention in the results that there was no significant difference in detection of either N1 or N2 from grab vs. composite sampling techniques. We have modified the text to read as follows:

“There was no significant difference between grab or composite sampling for the detection of either of the targets N1, N2...”

This article addresses an important topic in public health surveillance, consideration of environmental factors in wastewater analysis. **The paper understandable, however it would be better if the results and discussion were separate sections.**

Response 2.2: We thank the reviewer for this suggestion and have separated the presented information into two separate sections, which is in-line with the formatting requirements of *Microbiology Spectrum*.

Authors here discussed results showing significantly higher quantifications for N2 versus N1. Other studies have shown this may be due to mutations in SARs-CoV-2. This should be discussed/mentioned in the current study.

Response 2.3: We have cited the work by Endo et al. discussing how qPCR underestimated N1 copies after July 2022 in the discussion.

Reviewers also stated that composite samples showed significantly higher delta copies while grab samples showed significantly higher omicron copies. Do researchers have a theory/explanation for this? Were the composite samples and grab samples taken during the same time frame? Is it possible that as time passed one variant became more dominant, thus skewing rates?

Response 2.4: We apologize for the confusion, all of the samples were collected between March 2021 and April 2022 and any results discussing differences between grab and composite samples were collected between August 2021 and April 2022, after the composite samplers were deployed. The composite and grab samples were taken from the same locations on the same day, with the composite samplers deployed 24 hours prior and the same archived samples were tested for SARS-CoV-2, Delta, and Omicron. We have added timelines to the supplemental figures and have referenced them in the text to clarify these points for readers.

Did authors detect differences between semesters? As students left for school breaks, it's possible there was more exposure to one variant over another, particularly as time went on.

Response 2.5: We thank the reviewer for this insight and have added a timeline to the supplemental figures (Fig S1) with the school breaks indicated.

Re: Spectrum02003-24R1 (The influence of environmental factors on the detection and quantification of SARS-CoV-2 Variants in Dormitory Wastewater at a Primarily Undergraduate Institution)

Dear Dr. Rachel Bleich:

Thank you for your substantially revision manuscript describing a large dataset of SARS-CoV-2 wastewater surveillance. The inclusion of a variety of metadata greatly enrich the utility of the abundance measurements.

Your manuscript has been accepted, and I am forwarding it to the ASM production staff for publication. Your paper will first be checked to make sure all elements meet the technical requirements. ASM staff will contact you if anything needs to be revised before copyediting and production can begin. Otherwise, you will be notified when your proofs are ready to be viewed.

Sincerely,
JJ Miranda
Editor
Microbiology Spectrum